# Sensitivity Improvement and Determination of Rydberg Atom-Based Microwave Sensor

**Minghao Cai [1,2]** , **Zishan Xu [1,2]**, **Shuhang You [1,2]** and **Hongping Liu [1,2],***

[1]  State Key Laboratory of Magnetic Resonance and Atomic and Molecular Physics, WIPM, Innovation Academy for Precision Measurement Science and Technology, Chinese Academy of Sciences, Wuhan 430071, China; minghaoc@126.com (M.C.); xuzsxzis@126.com (Z.X.); youshuhang@outlook.com (S.Y.)

[2]  University of Chinese Academy of Sciences, Beijing 100049, China

*  Correspondence: liuhongping@apm.ac.cn

**Abstract:** We present a theoretical and experimental investigation of the improvement and determination of the sensitivity of Rydberg atom-based microwave RF sensor. An optical Bloch equation has been set up based on the configuration that two-color cascading lasers exciting atom to highly Rydberg state and a microwave RF coupling this Rydberg state to its adjacent neighbor. The numerical simulation shows that the sensitivity of the atomic RF sensor is correlated with the amplitude strengths of the applied two lasers and the RF itself. It also depends on the frequency detuning of the coupling laser, which induces an asymmetrically optical splitting. The coupling laser frequency fixing at the shoulder of the stronger one is more favorable for a higher sensitivity. Accordingly, we perform an experimental demonstration for the optimization of all these parameters and the sensitivity is improved to 12.50(04) nVcm$^{-1}$·Hz$^{-1/2}$.

**Keywords:** microwave transition; Rydberg atoms; electric dipole moment





## 1. Introduction

Rydberg atoms in highly excited states with one or more electrons of large principal quantum numbers are sensitive to electric fields, very suitable to manufacture atom-based sensor for detecting and receiving communication signals [1]. It has been widely investigated thoroughly both theoretically and experimentally throughout the last decades [2–13]. This type of sensors can replace the front-end components and electronics in a conventional antenna/receiver system [14,15], since they have potential advantages over conventional systems. This Rydberg-atom based quantum sensor owns unique properties such as self-calibration and fine spatial resolution in both the far-field and near-field [14,16–18].

The spacing between Rydberg levels can locate across the microwave radio frequency, which can be used to measure RF E-field strengths over a large range of frequencies (1 GHz to 500 GHz) with a high sensitivity [5,19,20], approximately < 1 µVcm$^{-1}$·Hz$^{-1/2}$ [21,22]. Various groups have investigated wireless communication using Rydberg atoms [2,7,23–25]. For example, utilizing the Rydberg atoms as a receiving antenna, Deb et al. directly recovered signal at the baseband without any demodulation means [2]. Anderson et al. demonstrated an atomic receiver for amplitude modulation (AM) and frequency modulation (FM) communication with a 3 dB bandwidth in the baseband of 100 kHz [23]. All of their work show atom-based quantum techniques are new and promising candidates for microwave communication applications.

An RF E-field applied to the Rydberg atoms results in the Autler-Townes (AT) splitting [26] of a ladder-type Rydberg electromagnetic-induced transparency (EIT) transmission spectrum [27]. The measurement of the RF E-field with Rydberg atoms can be converted into frequency measurement. The AT splitting ($\Delta f$) is proportional to the RF E-field strength $E$ as described as [14]

$$2\pi\Delta f = \Omega_{RF} = \frac{\mu_d E}{\hbar}, \tag{1}$$

and inversely the RF E-field strength is expressed as

$$E = \frac{2\pi\hbar}{\mu_d}\Delta f,$$ (2)

which quantitatively associates the RF E-field strength and the AT spectral splitting, where $\Omega_{RF}$ is the Rabi frequency of the Rydberg state transition induced by the RF E-field and $\mu_d$ is the transition dipole moment of the adjacent Rydberg states and can be calculated accurately, and $\hbar$ is Planck's constant. We can see that a larger dipole moment $\mu_d$ corresponds with a larger gain coefficient to magnify a weak electric field observable by the optical AT-splitting. If the coupling laser is scanned, the AT-splitting $\Delta f$ is exactly equal to the spectral splitting in frequency domain.

When we apply amplitude modulation (AM) to the RF carrier, probe transmission also carries a relating modulation signal and can be directly measured using a fast photo-diode detector. Owing to a potential high sensitivity, the quantum receiver can be used for very weak signal communication, which can also greatly reduce the cost of a transceiver system [15,28].

As the MW sensing is via the RF-optical conversion, principally, it is possible to improve the sensitivity of electric field measurements via optimizing the probe beam and coupling beam power [29,30], local electric field strength and frequency detuning [9,31] together. It is also vital to evaluate the individual contribution of each physical parameter in enhancing the MW detection sensitivity. In this work, we study all the parameter optimizations as well as the resonant transmission point of the local electric field, which is expected to give a better sensitivity.

## 2. Theory and Experimental Setup

A typical schematic of the energy levels of $^{85}$Rb based on the two-color-laser excitation [9,29–31] is shown in Figure 1 as well as the experimental realization. Theoretically two different laser systems are needed to address the Rydberg levels in Rb, one near 780 nm and one near 480 nm. The 780 nm light is tuned to the $D_2$ transition in Rb commonly, corresponding to the transition $|5^2S_{1/2}, F = 3\rangle \rightarrow |5^2P_{3/2}, F = 3\rangle$, and the 480 nm one continues to excite the atom on $|5^2P_{3/2}\rangle$ to high Rydberg state $|70^2S_{1/2}\rangle$, forming a cascading irradiation configuration. The microwave RF couples the Rydberg state to its neighbor partner $|70^2P_{3/2}\rangle$ with its electric dipole interaction strength monitored by the transparency of the probe beam of 780 nm through the atomic vapor.

The dynamics of interaction between lasers (RF) and atom is governed by the following time-dependent optical Bloch equation [32,33]:

$$\frac{d\rho(t)}{dt} = -\frac{i}{\hbar}[H(t), \rho(t)],$$ (3)

where $\rho(t)$ is the density matrix in interaction picture, and $H(t)$ is the Hamiltonian in a complex form so as to include all relaxations of atom levels [34]. Note the exception that only the diagonal imaginary part follows the operation of classical brackets.

Taking the energy level of the intermediate state $5^2P_{3/2}$ as a reference, the Hamiltonian $H$ can be expressed as a matrix form

$$H = \begin{bmatrix} \Delta_c + \Delta_{RF} - i\frac{\gamma_{RF}}{2} & -\frac{\Omega_M}{2} & 0 & 0 \\ -\frac{\Omega_M}{2} & \Delta_c - i\frac{\gamma_c}{2} & -\frac{\Omega_c}{2} & 0 \\ 0 & -\frac{\Omega_c}{2} & -i\frac{\gamma_p}{2} & -\frac{\Omega_p}{2} \\ 0 & 0 & -\frac{\Omega_p}{2} & -\Delta_p \end{bmatrix},$$ (4)

where a probe laser couples the ground state $5S_{1/2}$ and intermediate state $5P_{3/2}$, characterized by its Rabi frequency $\Omega_p$ while a coupling laser $\Omega_c$ induces an interaction between the intermediate state $5P_{3/2}$ and the Rydberg state $70S_{1/2}$. Different from the usual Ladder-type

EIT configuration, a microwave field is additionally applied and correlates this Rydberg state to another close neighbor Rybderg state $70P_{3/2}$ with large Rabi frequency $\Omega_{RF}$ due to the considerable dipole moment between Rydberg states. The equation $\Delta_i$ ($i = $ p, c, RF) depicts the frequency detunings of the probe, coupling beams and the microwave, respectively, while $\gamma_i$ ($i = $ p, c, RF) stands for the natural linewidth of the intermediate and two Rydberg states. The instantaneous steady-state density matrix component $\rho$ associated with the transition corresponding to the probe beam can be averaged by weighting the Doppler effect to give a simulation comparable to experiment [35].

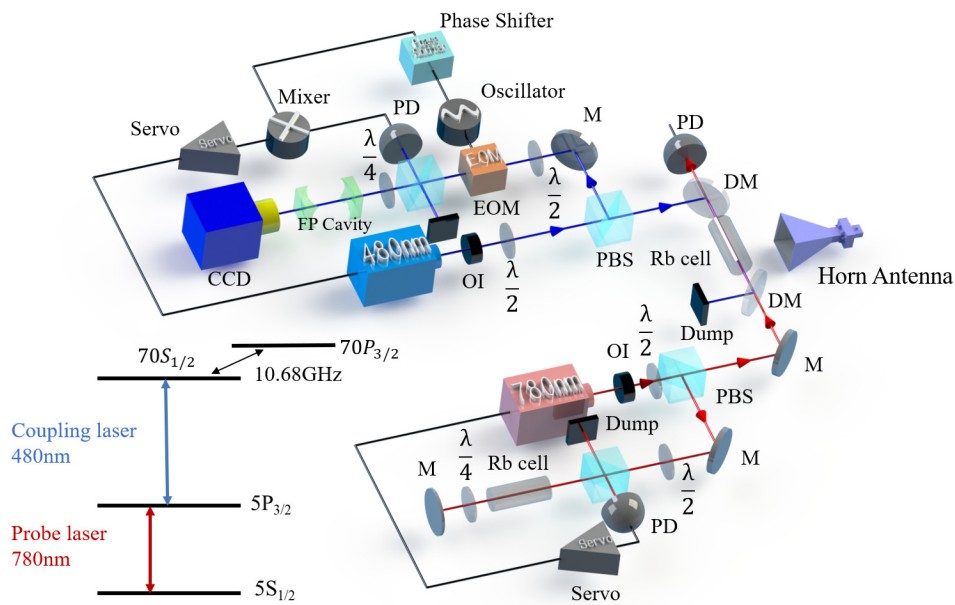

**Figure 1.** Experimental setup and its concerned four-energy-level diagram. The coupling ($\lambda_c = 480$ nm) and probe ($\lambda_p = 780$ nm) beams counter-propagate through a Rb vapor cell, forming a ladder-type EIT with upper Rydberg states coupled further by microwave RF. It is emitted by a horn antenna, serving as a local RF electric field oscillator (LO). The probe beam passing through an Rb cell and a dichroic mirror (DM) is detected by a photodiode (PD). The probe beam is frequency-locked to $|5^2S_{1/2}, F = 3\rangle \rightarrow |5^2P_{3/2}, F = 3\rangle$ transition via an integrated saturated absorption spectroscopy (SAS) unit of $^{85}$Rb atoms. The coupling beam scans across transition $|5^2P_{3/2}\rangle \rightarrow |70^2S_{1/2}\rangle$ and can also be frequency-locked to this transition via a Pound-Drever-Hall technique (PDH) based on an ultra-stable cavity on demand.

A theoretical simulation of the probe transmittance by scanning the coupling laser based on the optical Bloch equation is presented in Figure 2. We can see that the transmittance reduces a little with the applied RF strength increasing from zero to $\Omega_{RF} = 2\pi \times 4$ MHz but the spectral line doesn't split. After this point, the transmittance at zero detuning of the coupling laser increases quickly as the spectral line gets split. It is shown in Figure 2a. The probe transmittance dependent on the RF strength at zero detuning and its derivative depicting its RF-optical gain are shown in Figure 2b,c. It approximately follows a linear relationship within the RF strength, i.e., between $2\pi \times 4$ MHz and $2\pi \times 14$ MHz, especially at point A in Figure 2b or the point B in Figure 2c, where the curve has a largest slope (Figure 2b), corresponding to an RF strength $2\pi \times 7$ MHz. It implies that the atom has a maximum sensitivity for RF change at this critical value, providing a reference for the experiment [22]. This point has another merit that it is very close to the RF-strength at which the AT-splitting begins to occur [4]. It corresponds an RF electric field in scale of $E_L \sim 3.0$ mV·cm$^{-1}$ [22]. Around this point, it thus has a more direct self-calibration for the electric field strength of the RF according to Equation (1). The parameters of $\Delta_p = 0$, $\gamma_{RF} = \gamma_c = 2\pi \times 10$ kHz, $\gamma_p = 2\pi \times 6$ MHz, $\Omega_c = 2\pi \times 0.69$ MHz, $\Omega_p = 2\pi \times 11.3$ MHz and a room-temperature of 24 °C have been used in the simulation.

To avoid the Doppler background in the probe transparent spectrum, we record the spectrum by scanning the coupling laser frequency of 480 nm while the probe laser of 780 nm locked to a saturation absorption spectroscopy (SAS) of Rb. The probe beam is derived from a cat-eye Morglab laser at 780.246 nm with maximum output of 180 mW and a free running linewidth of ~1 MHz. The coupling laser (480 nm) is generated by a frequency-doubled amplified diode laser system (TA-SHGpro, Toptica, Munich, Germany), which is scanned across the $|5P_{3/2}\rangle \rightarrow |70S_{1/2}\rangle$ hyperfine transition and real-time monitored by an FP-cavity for relative frequency reference. Both lasers can also be selectively locked to an ultralow expansion glass (ULE) Fabry-Perot (FP) cavity using the Pound-Drever-Hall technique (PDH) with two pairs of mirrors mounted on the same supporting skeleton immune to the room temperature fluctuation, within less than 1 kHz. The standard PDH-locking skills have been used in the feedback loop where the laser beams imposed to the FP-cavity are modulated by an electro-optical modulator (EOM) driven by a local oscillator of ~20 MHz. The locking signal comes from the mixing between the phase-shifted local oscillator and the photo-diode signal of laser reflected from the FP cavity, respectively. The 780 nm beam is focused to a spot of waist 750 μm with typical power of 100 μW, corresponding to Rabi frequency $\Omega_p = 2\pi \times 11.13$ MHz. While the 480 nm beam focused to 1.25 mm with typical power of 420 mW, corresponding to Rabi frequency $\Omega_c = 2\pi \times 0.69$ MHz. They have already been used in the theoretical simulation.

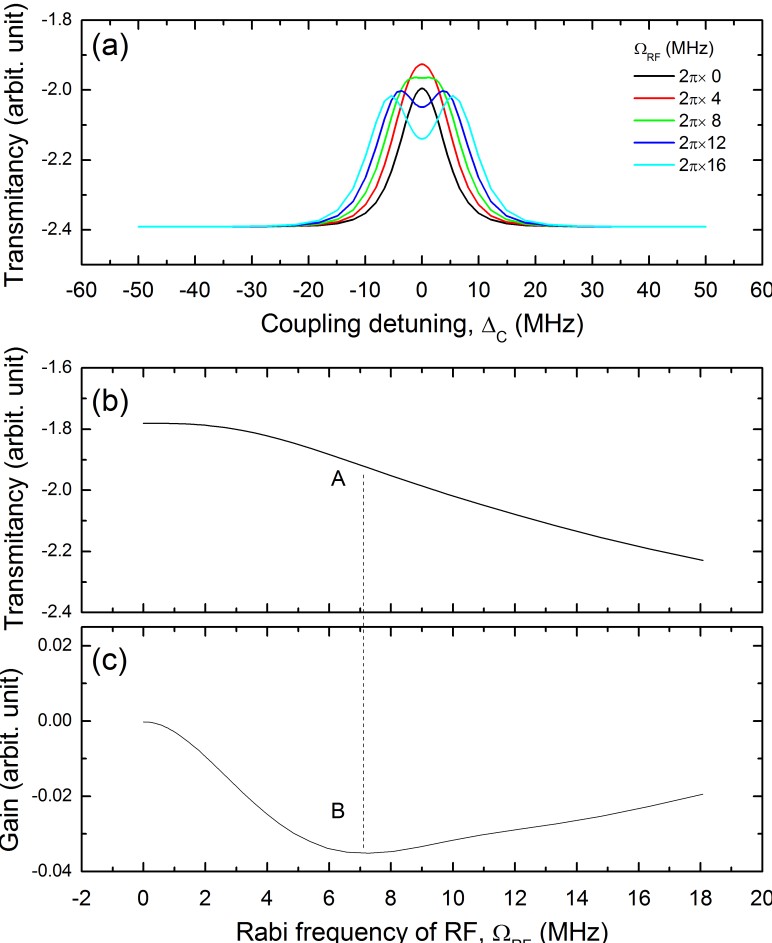

**Figure 2.** (**a**) Theoretical simulation of the probe transmittancy by scanning the coupling laser based on the optical Bloch equation at various RF strengths, (**b**) the probe transmittancy dependent on the RF strength at zero coupling detuning and its derivative depicting its RF-optical gain. The point A in (**b**) and B in (**c**) corresponds to the maximum gain where the RF strength is $2\pi \times 7$ MHz.

Based on the optical EIT scheme that the coupling ($\lambda_c = 480$ nm) and probe ($\lambda_p = 780$ nm) beams counter-propagate through an Rb vapor cell of length 70 mm and diameter 25 mm, a horn antenna has been used to generate a local RF E-field oscillator (LO) to couple the upper two Rydberg states $|70^2S_{1/2}\rangle$ and $|70^2P_{3/2}\rangle$ with frequency $\sim 10.68$ GHz in free-space. It corresponds to the area close to the point A in Figure 2b, which will be discussed later. The transition dipole moment is calculated from the ARC library [36] and its value is determined as $\mu_d = 2395$ a.u.

## 3. Results

As demonstrated in the simulation, the probe optical transmission response is not linearly dependent on MW RF powers and the LO electric field strength should be chosen at the point with maximum slope [22]. At lower RF electric field, the AT-splitting is very small and the probe optical response to the applied MW power is not sensitive [37]. On the contrary, at very high RF carrier power, the AT-splitting is large enough where the splitting lines have already been separated. At this moment, the probe optical response is not sensitive, either. An intermediate case is preferred at which point the optical response has the largest slope on the applied RF power. This point also relies on the power of probe and coupling laser powers.

We present the measurement of the optical response on the power of RF generator in Figure 3a at different probe laser powers. We can see that a higher power of probe laser can bring larger optical transmission signal and also larger slope or response gain for RF power. We can see that the best gain locates at the point with maximum slope, around $-13$ dBm, corresponding to Rabi frequency of 6.97(13) MHz, consistent with the theoretical simulation in Figure 2b,c . However, we choose the optimized RF power at $-10$ dBm (corresponding to an electric field of 3.53(18) mV·cm$^{-1}$ at the vapor cell) as higher RF power makes the AT-splitting more obvious and results in a more direct and thus more accurate RF-field calibration. Moreover, the gains within this area are very close and a little loss of sensitivity is worthy for reliable calibration.

We further investigate the optical response dependent on the probe laser intensity. It is shown in Figure 3b, where the response increases linearly with the probe laser intensity. However, unfortunately, the AT-spectral line width increases, either, nearly linearly. Specially, at a power of 300 µW, the probe laser is too strong, leading to a spectral splitting. We should balance these two factors in a real experimental operation. As can be seen later, we prefer a value of 100 µW so as to keep narrower linewidth and to keep far away from the critical point of splitting.

The optimization also relies on the coupling laser power. We present the measurement of the optical response on the power of RF generator in Figure 4a at different coupling laser powers. Similar to the case of probe laser power optimization, a higher power of laser can bring larger optical transmission signal and also larger response gain for RF power. At the optimized RF power of $-10$ dBm, the optical response dependent on the coupling laser intensity also shows a linear relation. It is shown in Figure 4b. Unlike the case of optimization of probe laser, however, the spectral linewidth remains almost unchanged (Figure 4c). It implies we can choose a higher coupling laser power if possible. It is also physically reasonable since a stronger coupling of 480 nm laser is more favorable for the transfer of RF AT-splitting information down to the probe laser.

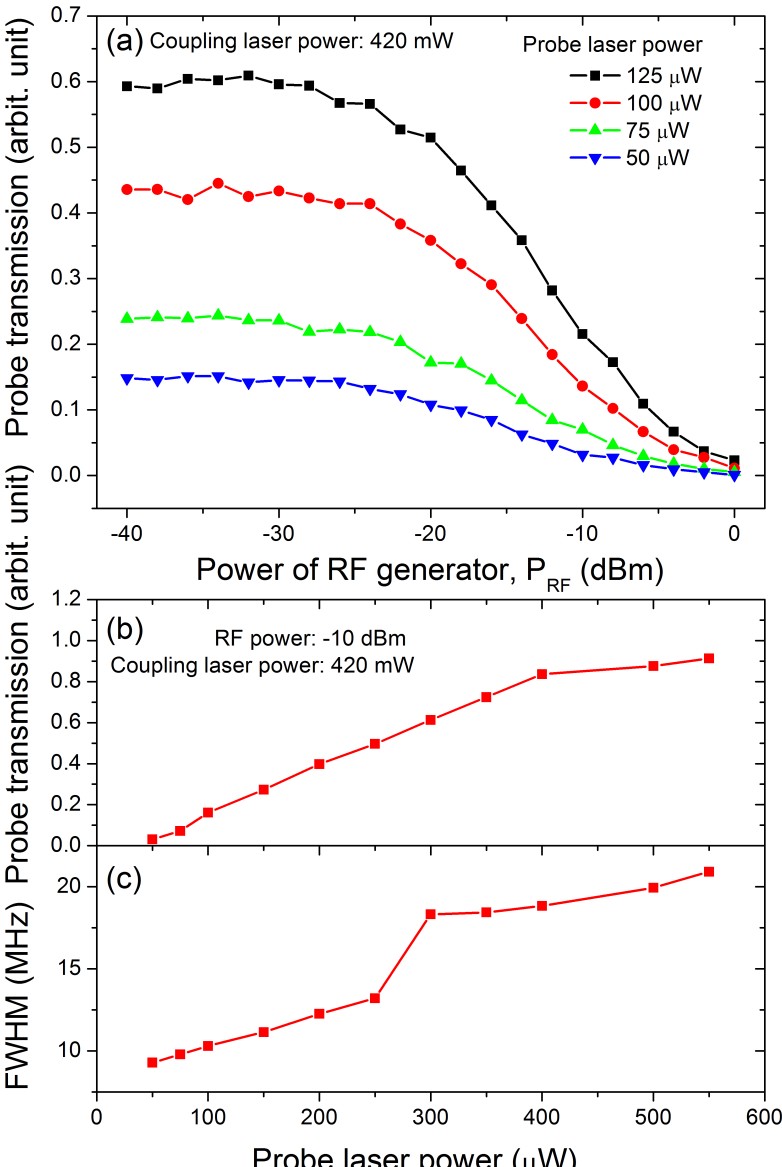

**Figure 3.** (**a**) Probe optical transmission response on the applied RF power at various probe laser powers. (**b**) At the optimized RF power of −10 dBm and coupling laser intensity of 420 mW, the optical signal increases linearly with the probe laser intensity, but (**c**) the AT-spectral linewidth also almost linearly increases. An optimized probe laser power should keep a balance between them.

As we have simulated previously in Figure 2, the antennas have the highest dynamic response to RF changing and then more direct self-calibration for the RF electric field at point close to AT-splitting [22]. Typical AT-splittings at various RF power and one of their dynamic responses are shown in Figure 5a. The RF E-field couples a pair of specific Rydberg states, $|70^2S_{1/2}\rangle$ and $|70^2P_{3/2}\rangle$. The typical AT-splittings are measured at RF power −12, −10 and −8 dBm, corresponding to AT-spectral line separation of 7.94(9), 11.48(9) and 13.75(10) MHz. We can deduce the electric field strength of the local RF field according to Equation (2). Thus we can measure a series of data points of RF E-field versus the applied RF power in the vicinity of a given LO power, for example, at the point corresponding to E-field of 3.53(18) mV·cm$^{-1}$. The data are shown in black squares in the Figure 5b.

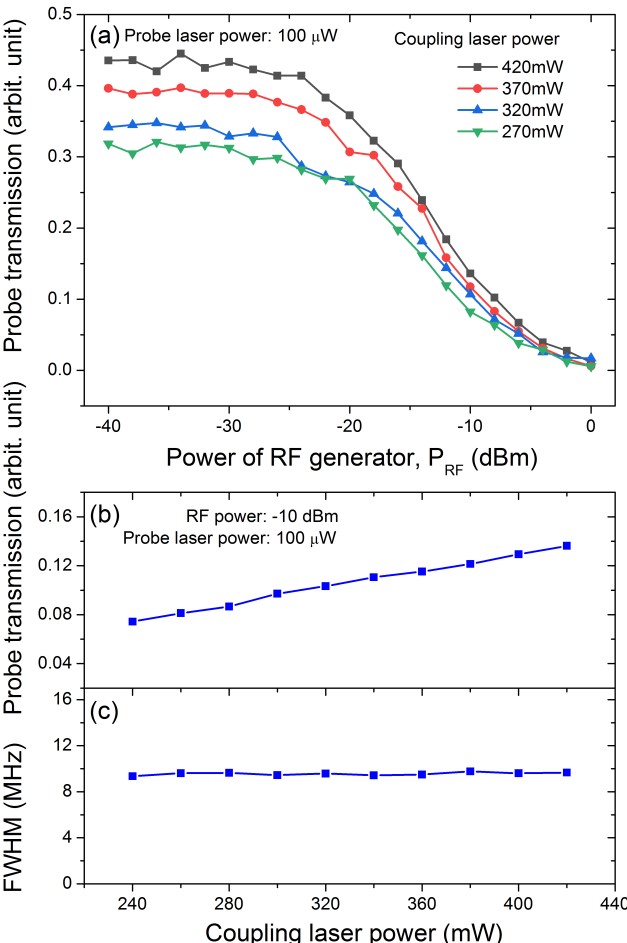

**Figure 4.** (**a**) Probe optical transmission response on the applied RF power at various coupling laser powers. (**b**) At the optimized RF power of $-10$ dBm and probe laser intensity of 100 μW, the optical signal increases linearly with coupling laser intensity, while (**c**) the AT-spectral linewidth nearly unchanged.

As the RF E-field is proportional to the square root of the output power, given by the standard antenna equation $E_L = \sqrt{\eta\alpha_1 g P_{RF}/4\pi d^2}$ [22], where $P_{RF}$ is the power of the microwave source, $\eta$ is the intrinsic impedance of free space, $\alpha_1$ is the insertion loss between the RF source and antenna, $g$ is a gain of the antenna, and $d$ is a distance from the antenna to the cell. Considering $\eta$, $\alpha_1$, $g$ and $d$ are constants for a certain experiment, the relation can be simplified as $E_L = \alpha\sqrt{P_{RF}}$, where $\alpha = \sqrt{\eta\alpha_1 g/4\pi d^2}$ is the effective gain of horn antenna. In our experiment, the linear least square fit determines the effective gain as $\alpha = 43.16(64)$ Vm$^{-1}\cdot$W$^{-1/2}$. The fit line is plotted in red solid line in Figure 5b. The quality of the linearity of the measurements is also very good, excluding the complex state interaction induced nonlinearity [38]. Therefore, the weak RF E-field value can be obtained by the above formula at given antenna power.

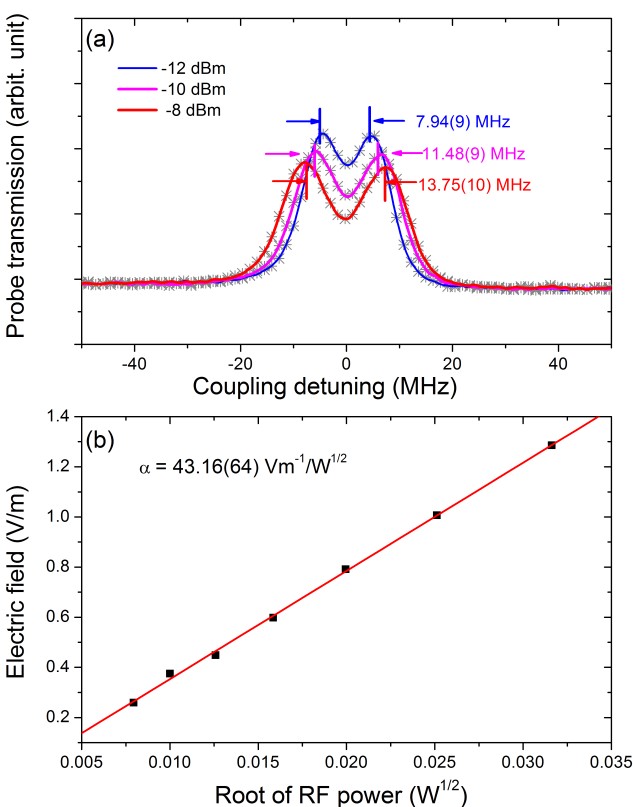

**Figure 5.** (**a**) Typical AT-splittings due to the RF induced interaction between the adjacent Rydberg states at different RF powers. (**b**) Measurements of the local RF E-field $E_L$ (black square) by EIT-AT spectra versus the square root of the output power, $\sqrt{P_{RF}}$, and its linear fit (red line) by formula $E_L = \alpha \sqrt{P_{RF}}$ with parameter determined as $\alpha = 43.16(64)$ Vm$^{-1}$/W$^{1/2}$.

We can also turn back to the optimization of the MW RF power according to the dynamical response by applying a weak amplitude modulation on the RF generator. The measurement of optical response of MW at different RF powers varying from $-14$ to $-4$ dBm is shown in Figure 6(a1,b1,c1,d1,e1). To see the dynamical response more clearly we extract the dynamical signal by filtering the flat DC part away, which is shown in Figure 6(a2,b2,c2,d2,e2). We can see that when the RF power increases up $-10$ dBm, the response has a large amplitude and nearly gets saturated, implying an optimized RF power. The optimized RF power corresponds to an electric field intensity of 0.353(18) V/m, close to the value of 0.3 V/m in Ref. [22]. The above measurement is performed at laser power fixed at the optimized intensity in previous steps shown in Figures 3 and 4. The amplitude modulation frequency is taken as 2 kHz with magnitude of 0.5 V at the input interface of the RF microwave generator, corresponding to a maximum electric field of 1.720(25) mV·cm$^{-1}$ at the vapor cell.

To further improve the sensitivity of atomic antenna in MW communication, we detune the carrier MW frequency off resonance forming an asymmetrically splitting optical response, which is similar to the case of Cs atom [9]. There the MW is detuned from the resonant transition between two Rydberg states, $47D_{5/2} \leftrightarrow 48P_{3/2}$, corresponding to a value of 6.946 GHz. At this moment, we can compare the optical response of amplitude-modulated signal MW RF at a given local RF power but at resonance and off-resonance to the transition between the adjacent Rydberg levels. It is shown in Figure 7. As the same as in Figure 6, the amplitude modulation frequency is taken as 2 kHz with magnitude of 0.5 V at the input interface of the microwave generator. An intuitive view of Figure 7 tells us that a moderate RF detuning can help to increase the dynamical signal, especially at stronger RF power $-10$ dBm as shown in Figure 7b and $-4$ dBm in Figure 7c.

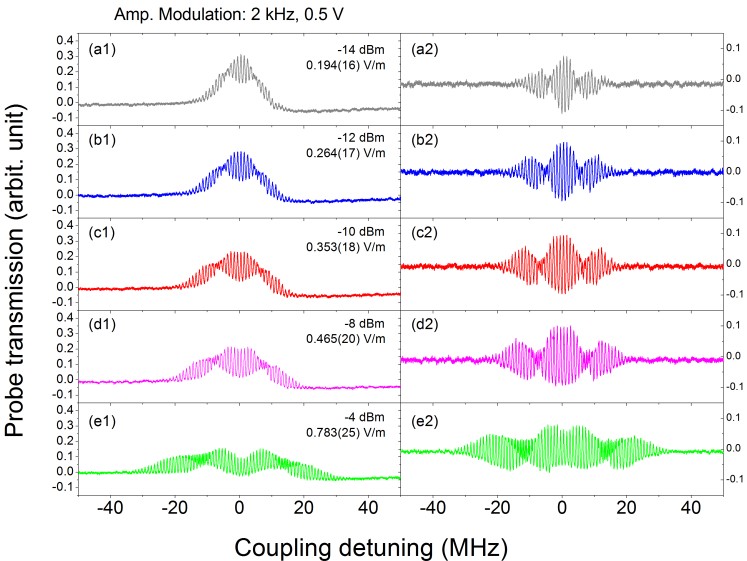

**Figure 6.** The measurement of optical response of MW at different RF powers varying from −14 to −4 dBm (**a1,b1,c1,d1,e1**) and the dynamical signal extraction (**a2,b2,c2,d2,e2**). The laser powers are fixed at the optimized intensities. When the RF power increases up −10 dBm, the response has a large amplitude and nearly gets saturated, implying an optimized RF power.

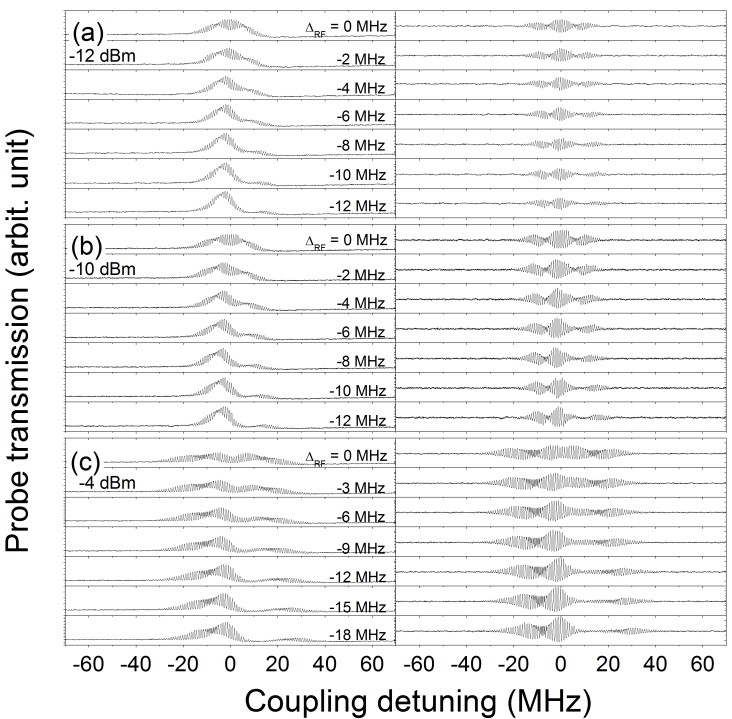

**Figure 7.** The measurement of optical response of MW at different RF frequency detunings (**left**) and its dynamical signal extraction (**right**) at various LO field power (a) -12 dBm, (b) -10 dBm and (c) -4 dBm. A moderate detuning can increase the dynamical signal. The laser powers are fixed at the optimized intensities.

For a better view of the result in Figure 7, we can extract the maximum dynamical signal amplitude and redraw it in Figure 8. We can see that at RF power −12 dBm, the optimization is located at detuning $\Delta_{RF} = -6$ MHz but at higher RF power, −10 dBm and −4 dBm, the better LO power goes up to larger detuning values, −10 and −27 MHz, respectively. Specially, at RF power of −10 dBm, the dynamical signal is not sensitive to

the RF detuning in a more wide range, −20 to −6 MHz. This feature is more useful in real experimental application.

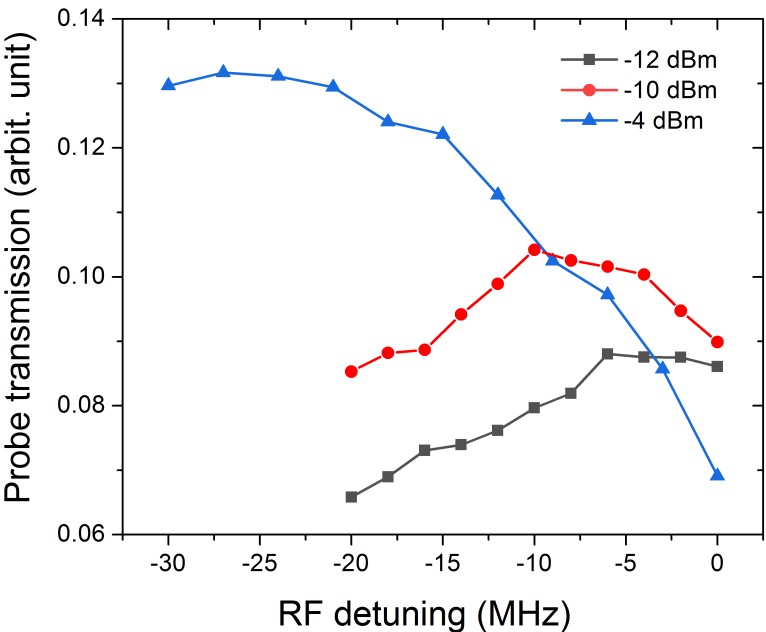

**Figure 8.** The maximum dynamical signal response dependent on the RF detunings at different MW RF powers in Figure 6, where obvious optimization can be determined.

The enhancement of dynamic optical response by RF detuning can be understood by a theoretical simulation based on the optical Bloch equation considering the four energy levels [10,11,13,22,38,39] as detailed previously. The simulation is shown in Figure 9. Theoretical simulations of the optical transparency response for the RF detuning are presented in Figure 9a,b, corresponding to RF detuning of $\Delta_{RF} = 0$ and −6 MHz. A small amplitude-modulation has been applied to the RF field under the modulation. From the above spectra, we can obtain the corresponding dynamic optical responses as Figure 9c,d. We can see that the strong dynamical response locates at the shoulder of the transparency spectral line rather than the spectral peak itself. Moreover, a little enhancement of the dynamical response can be achieved by RF detuning of −6 MHz at Figure 9d. The RF detuning causes the break-down of the symmetry of the optical response and transfers the dynamic optical gain to the branch closer to the zero detuning. There still remain some discrepancies with the experiment due to the inaccurate determination of laser-atom coupling strength.

Finally, we quantitatively evaluate the sensitivity of the Rydberg RF sensing system. Here 1 kHz modulation is applied for better RF-optical response since the gain begins to reduce with modulation frequency increasing, especially damp quickly at larger than 2 kHz. Following the theoretical simulation, a small signal (SIG) produced by a RIGOL function waveform generator has already been amplitude-modulated to the RF LO to visualize the signal detection and amplification. The applied amplitude modulation intensity decreases until approaching the lowest visual perceptible threshold at RF frequency resonant to and detuned away from the transition between the two adjacent Rydberg states. The optical dynamical response to these two frequencies are shown in Figure 10 where all other parameters fixed at the optimized values. The LO RF power is set to −10 dBm as previously determined. Figure 10a,b correspond to the case of RF resonance while Figure 10a′,b′ to the case at the optimized detuning of −6 MHz. The large modulation gives a nice sinusoidal fit for the measured data points as presented in Figure 10a,a′, corresponding to a signal RF of 860(13) μV/cm.

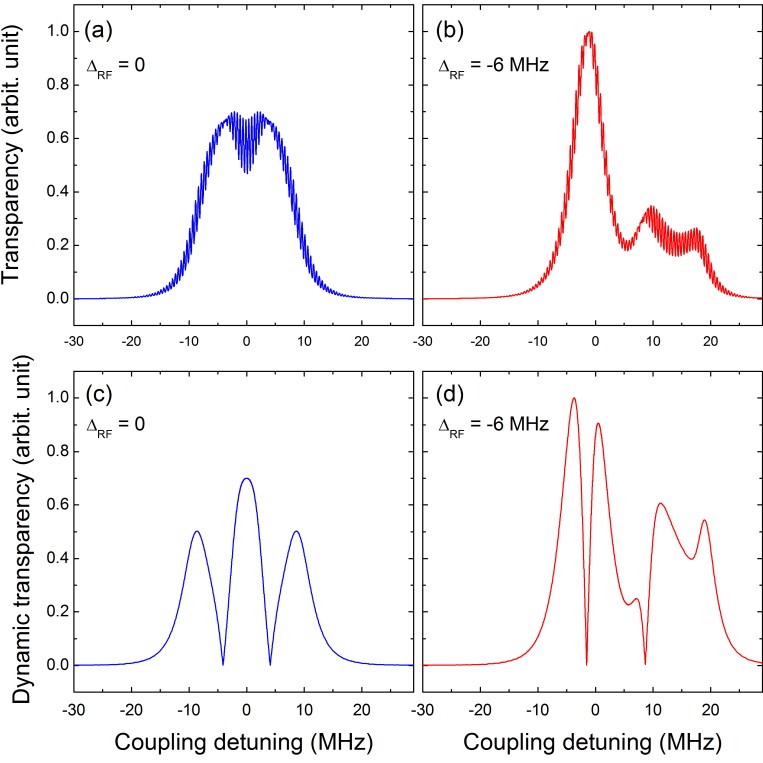

**Figure 9.** Theoretical simulation of the dynamical optical response for the RF detuning. (**a,b**) are the probe transparency signal under the 1 kHz modulation at zero and −6 MHz RF detuning, respectively. Their dynamic optical response extractions are shown in (**c,d**), where a little increase in signal can be obtained by RF detuning.

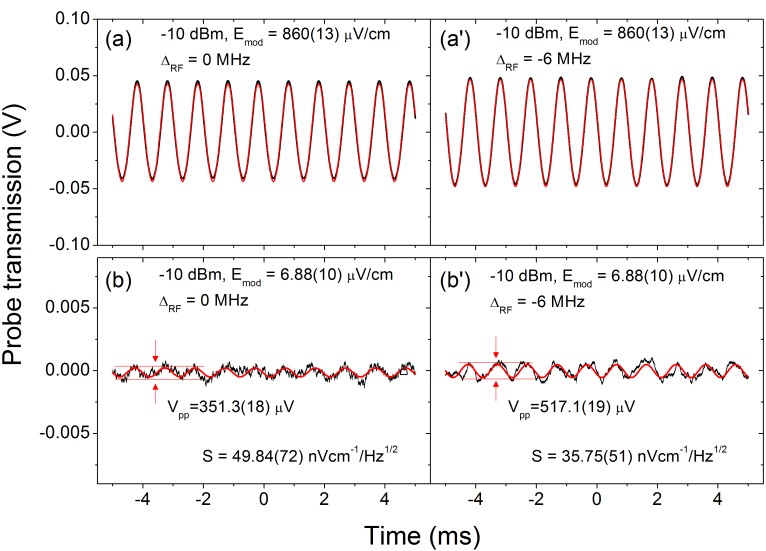

**Figure 10.** The optical dynamical response at two RF detuning frequencies but all other parameters fixed at the optimized values. The applied amplitude modulation intensity decreases from (**a**) $E_{mod} = 860(13)$ μV/cm down to (**b**) $E_{mod} = 6.88(10)$ μV/cm at LO RF field resonance until approaching the lowest visual perceptible threshold. (**a′,b′**) correspond to the case of LO RF detuning of −6 MHz.

It should be noted that the modulation is also calibrated by AT-splitting according to Equation (2) by setting a bias DC voltage on the input port of the RF-generator since it is DC-coupled. It proves the modulation intensity is proportional to the applied voltage. Thus the RF-optical gain $\beta$ shown in Figure 2b,c can be determined, $1/\beta = 13,312(47)$ μVcm$^{-1}$/V. For example, the voltage $V_{pp} = 351.3(18)$ μV corresponds

to RF-field of $E_{mod} = 6.88(10)$ µV/cm in Figure 10b. In the detuning case, a higher voltage of $V_{pp} = 517.1(19)$ µV is required to correspond to RF-field of $E_{mod} = 6.88(10)$ µV/cm in Figure 10b, indicating a better sensitivity.

When the modulation deceases to a lower perceptible threshold, corresponding to $6.88(10)$ µV/cm at the vapor cell, the detuning method has a remarkably better sensitivity, indicating a sensitivity improvement. The latter optical response gives a high value of $517.1(19)$ µV on the photo detector, larger than that of $351.3(18)$ µV at zero detuning. The method without detuning gives a sensitivity of $49.84(72)$ nVcm$^{-1}$·Hz$^{-1/2}$ while the detuning of $-6$ MHz gives a sensitivity of $35.75(51)$ nVcm$^{-1}$·Hz$^{-1/2}$.

A further step is to measure the noise spectrum with zero and $-6$ MHz detuning. The RF detuning of $-6$ MHz provides a sensitivity of $12.50(04)$ nVcm$^{-1}$·Hz$^{-1/2}$, better than that $13.69(40)$ nVcm$^{-1}$·Hz$^{-1/2}$ at zero detuning as well. In the estimation, the sensitivity is achieved by formula $S = E_{min}\sqrt{T}$, where $E_{min} = 8.839(31)$ nVcm$^{-1}$ is obtained by fitting the noise spectrum within acquisition time $T = 2$ s, where the RF-optical gain parameter $\beta$ is used to transfer the optical amplitude to RF-field strength. It corresponds to a smallest discernible RF electric field of $176.78(57)$ pVcm$^{-1}$, measured at a time scale of 5000 s, slightly better than that $780$ pVcm$^{-1}$ with sensitivity of $55$ nVcm$^{-1}$·Hz$^{-1/2}$ in Ref. [22].

## 4. Conclusions

We have built up the highly excited Rydberg atom based RF field sensing system via electromagnetically induced transparency with two color cascading lasers. After choosing a higher Rydberg state as the sensing medium, we study the optical response of the probe beam dependent on the probe and coupling beam laser powers and fix them at the optimized values. A further optimization is performed on the LO RF power where the AT-splitting is the most sensitive to the dynamic RF electric field. At the same time, we detune the RF field frequency to break the symmetry of the AT-splitting. It leads to one of the asymmetric branches more sensitive to the RF electric field variation, which further enhances the RF field sensitivity. Based on the Bloch equation with four energy levels concerned, we theoretically simulate the dynamic optical response of the system, completely supporting the optimization mechanism.

**Author Contributions:** Conceptualization, H.L.; methodology, H.L.; software, Z.X.; validation, H.L.; formal analysis, S.Y.; investigation, M.C.; data curation, M.C.; writing—original draft preparation, M.C.; writing—review and editing, H.L.; project administration, H.L.; funding acquisition, H.L. All authors have read and agreed to the published version of the manuscript.

**Funding:** This work is supported by National Natural Science Foundation of China (NSFC) (No. 12074388 and No. 12004393).

**Institutional Review Board Statement:** Not applicable.

**Informed Consent Statement:** Not applicable.

**Data Availability Statement:** Not applicable.

**Conflicts of Interest:** The authors declare no conflict of interest.

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
