# Peer review of "Sensitivity Improvement and Determination of Rydberg Atom-Based Microwave Sensor"

_photonics, doi:10.3390/photonics9040250_

Round 1

Reviewer 1 Report

The Rydberg-atom-based microwave field measurement has been investigated widely in recent years, the MW detection sensitivity is pursued. The manuscript perform an experimental demonstration for the optimization of the probe and coupling beam laser powers, the RF power and the RF field frequency, the sensitivity is improved to 12.50(04) nVcm−1 · Hz−1/2 by the dynamical optical response for the RF detuning. Such result is slightly better than the best reported. However, due to lacking physical explanation of their results and relying only on theoretical model calculations, it seems this method is not universal. and lacking physical explanation of their results. The followings are my comments and questions

1,Theoretical models are very conventional in which have many approximations, and don't even take into account the Doppler effect. And the experiment uses room temperature atoms. In fact, for such a high Rydberg (n=70), the influence of neighboring energy levels should be considered as well as velocity-changing collisions.

  1. line 84:“ a room-temperature of 30â—¦C have been used in the simulation”. Why not measure the actual temperature experimentally? Does the atomic pool have a temperature control system?

  2. Please explain this contradiction: line 83 and 84 :"Ωc = 2π × 10 MHz, Ωp = 2π × 5 MHz and 83 a room-temperature of 30â—¦C have been used in the simulation" ,however: line 101 and 102:"While the 480 nm beam focused 101 to 1.25 mm with typical power of 420 mW, corresponding to Rabi frequency Ωc = 2π × 0.69 102 MHz. "

  3. Eq.3 applies to this situation that is the probe laser is scanned, and it is not precise in the actual measurement[The effect of the Doppler mismatch in microwave electrometry using Rydberg electromagnetically induced transparency and Autler-Townes splitting,http://iopscience.iop.org/page/acceptedmanuscripts]. Since the manuscript record the 85 spectrum by scanning the coupling laser frequency to avoid the Doppler background in the probe transparent spectrum. Eq.3 should be modified.

  1. some mistakes:line 164-165,"different RF powers varying from −16 to −8 dBm is shown in Fig.6(a1-e1). " However, Fig.6 Caption:"Figure 6. The measurement of optical response of MW at different RF powers varying from −14 to −4 dBm (a1-e1) "

Reviewer 2 Report

The comments are attached below.

Reviewer 3 Report

The authors investigate the sensitivity improvement. Their results show the sensitivity of Rydberg sensor can be improved by optimizing the lasers power and choosing the LO electric field strength reaching maximum slope. In addition, they use a detuned MW field to increase the dynamical signal. Also, their calculation can match their results well. After the optimization of all above parameters, they achieve a sensitivity is of 12.50(04) nVcm-1Hz-1/2, which is even better than that of in Ref.[22].

As the Rydberg atom-based WM sensor has widely applications, and the author present pretty good results, in my view, the manuscript could be published in Photonics with certain revisions. Here are my few comments and questions about this manuscript.

For their finally measurement in Fig.10, how do they transfer the sine modulation to the strength of field (860(13)μV/cm or 3.88(10) μV/cm)?

Similar question, how do they get the sensitivity value (49.84(72) nVcm-1 Hz-1/2 for zero detuning or 35.75(51) nVcm-1 Hz-1/2 for -6MHz detuning)? Could have some detail calculation in manuscript?

The authors mention the amplitude modulation frequency is taken as 1 kHz with magnitude of 0.5 V.  What’s the value of modulation depth, is it 100%?

In line 184, should remove -8dBm as it is not shown in fig.7.

Round 2

Reviewer 1 Report

I have read the response letter and the updated manuscript. Authors answer my all questions, and I think the manuscript is ready to publish.

Reviewer 2 Report

None.